# Retrieving Corn Canopy Leaf Area Index from Multitemporal Landsat Imagery and Terrestrial LiDAR Data

**Wei Su [1,2,*], Jianxi Huang [1,2], Desheng Liu [3] and Mingzheng Zhang [1,2]**

1   College of Land Science and Technology, China Agricultural University, No. 17 Qinghua East Road, Haidian District, Beijing 100083, China; jxhuang@cau.edu.cn (J.H.); zhangmingzheng@cau.edu.cn (M.Z.)
2   Key Laboratory of Remote Sensing for Agri-Hazards, Ministry of Agriculture, Beijing 100083, China
3   Department of Geography, The Ohio State University, Columbus, OH 43210, USA; liu.738@osu.edu
*   Correspondence: suwei@cau.edu.cn; Tel.: +86-010-6273-7855

**Abstract:** Leaf angle is a critical structural parameter for retrieving canopy leaf area index (LAI) using the PROSAIL model. However, the traditional method using default leaf angle distribution in the PROSAIL model does not capture the phenological dynamics of canopy growth. This study presents a LAI retrieval method for corn canopies using PROSAIL model with leaf angle distribution functions referred from terrestrial laser scanning points at four phenological stages during the growing season. Specifically, four inferred maximum-probability leaf angles were used in the Campbell ellipsoid leaf angle distribution function of PROSAIL. A Lookup table (LUT) is generated by running the PROSAIL model with inferred leaf angles, and the cost function is minimized to retrieve LAI. The results show that the leaf angle distribution functions are different for the corn plants at different phenological growing stages, and the incorporation of derived specific corn leaf angle distribution functions distribute the improvement of LAI retrieval using the PROSAIL model. This validation is done using in-situ LAI measurements and MODIS LAI in Baoding City, Hebei Province, China, and compared with the LAI retrieved using default leaf angle distribution function at the same time. The root-mean-square error (RMSE) between the retrieved LAI on 4 September 2014, using the modified PROSAIL model and the in-situ measured LAI was 0.31 $m^2/m^2$, with a strong and significant correlation ($R^2 = 0.82$, residual range = 0 to 0.6 $m^2/m^2$, p < 0.001). Comparatively, the accuracy of LAI retrieved results using default leaf angle distribution is lower, the RMSE of which is 0.56 with $R^2 = 0.76$ and residual range = 0 to 1.0 $m^2/m^2$, p < 0.001. This validation reveals that the introduction of inferred leaf angle distributions from TLS data points can improve the LAI retrieval accuracy using the PROSAIL model. Moreover, the comparisons of LAI retrieval results on 10 July, 26 July, 19 August and 4 September with default and inferred corn leaf angle distribution functions are all compared with MODIS LAI products in the whole study area. This validation reveals that improvement exists in a wide spatial range and temporal range. All the comparisons demonstrate the potential of the modified PROSAIL model for retrieving corn canopy LAI from Landsat imagery by inferring leaf orientation from terrestrial laser scanning data.

**Keywords:** leaf area index retrieval; leaf angle distribution function; PROSAIL model; terrestrial LiDAR; corn

## 1. Introduction

Leaf area index (LAI) is an important parameter that controls many physical and biological processes in vegetation canopies and therefore controls the resulting productivity [1]. By definition, LAI is measured as the total one-sided leaf area per unit ground surface area for flat broad leaves;

for non-flat leaves; it is half of the total light-intercepting area per unit ground surface area [1]. In precision agriculture, LAI has been used to monitor crop growth and development [2], estimate crop yield [3,4], map and classify crop vitality and yield [5], and to detect early crop stress [6]. Corn (*Zea mays*) is a productive C4 crop and LAI is a very important indicator for corn yield formation. Thus, the reliable estimation of LAI is an important tool for monitoring corn growth, estimating yield, cultivating corn, and breeding new cultivars.

There are currently two kinds of LAI estimation methods: direct measurements and indirect methods [6]. Compared with direct destructive sampling measurements, indirect estimation methods that use optical instruments and remote sensing images offer the advantage of rapid analysis for large areas, thereby overcoming the drawbacks of labor-intensive and time-consuming "direct" methods. The optical instruments include commercial canopy analysers (i.e., SunSCAN, AccuPAR, LAI-2000/LAI-2200, DEMON, PASTIS-57, digital hemspherical photography, etc.) and smartphone applications (i.e., PocketLAI and LAISmart) [7,8]. These methods are non-destructive, and are based on the statistical and probabilistic approach of foliar element distribution and arrangement within the crop canopy [9].

For the LAI retrieval using remote sensing images, there are two methods: empirical methods and physical models [10]. Empirical methods depend on the location, season, and plant attributes such as species, age, and density; therefore, they do not generalize well over large areas. In contrast, methods based on physical models are more attractive because they are robust over large areas and can be adjusted to account for a wide range of situations.

PROSAIL is a popular physical radiative transfer model for LAI estimation [11]. It was developed to couple the SAIL bidirectional canopy-reflectance model [12] and the PROSPECT model of leaf optical properties [13,14] in the 1990s. Many researchers have estimated canopy biophysical variables using the PROSAIL model in agriculture, forestry, environmental science, and ecology application [11]. However, there are three major problems for retrieving canopy biophysical parameters using PROSAIL: (1) the unknown foliage angle distribution, (2) the error because of the nonrandom spatial distribution of foliage, and (3) the contribution of the supporting non-photosynthetic material to radiation interception [15]. The non-random nature of foliage can be solved by applying simplifications such as a clumping index [6]. This article focuses on the problem of corn leaf angle estimation and identifying non-photosynthetic material of a corn canopy.

Leaf angle is important for estimating the bidirectional reflectance of plant canopies in remote sensing applications [16,17] and PROSAIL output is sensitive to leaf angle input [11]. Despite the importance of measuring the leaf angle distribution, only the Campbell ellipsoidal leaf angle distribution function [18] is commonly used to approximate this distribution by means of empirical expressions that have been parameterized for many species. For many species, including corn, the leaf angle distribution changes throughout the growth process. For corn, new leaves are more erect during the early growth stages but become more horizontal later. Therefore, to accurately retrieve corn LAI, it is necessary to describe the leaf angle distribution throughout the growth process. In the present study, our goal was to develop a model of corn leaf angle distribution suitable for use in the PROSAIL model. Light detection and ranging (LiDAR) technologies have enabled researchers to characterize the 3D canopy structure of corn and other crops, especially based on the high density of the point cloud produced by the LiDAR (Light Detection And Ranging) pulses during terrestrial laser scanning (TLS) [19,20]. Currently, there are two kinds of methods for deriving leaf angle distributions: computing normal vectors of neighboring TLS points [19,21,22] and computing the angle of leaf clusters classified from TLS data [23–25]. Zheng and Moskal developed the least square fitting method to compute the normal vectors of artificial tree leaves from TLS points for leaf orientation retrieval [19]. Bailey and Mahaffee calculate the normal vectors of broad-leafed black cottonwood tree and grapevine canopy for leaf angle probability density function estimation [21]. Vicari et al. calculate the normal vectors of leaves TLS points for four broadleaf tree species and estimated their leaf angle distributions [22]. Li et al. clustered the simulated tree leaves using density-based segmentation method for retrieving leaf angle

distributions [24]. Xu et al. segmented single-leaf point cloud for extracting leaf angle distributions for three tree species (*Ehretia macrophylla*, crape myrtle and *Fatsia japonica*) [25]. These researches all focus on the tree leaf angle distributions using TLS points, especially the broadleaf tree species. Furthermore, this promising new technology of TLS is used to estimate the corn leaf angle distribution in this study, which offers the possibility of improving accuracy of corn canopy LAI retrieval. Therefore, our goal was to retrieve the corn canopy LAI using the inferred leaf angle distributions in a form suitable for incorporation in the PROSAIL model, with data provided by terrestrial LiDAR. Specifically, we aim to (1) develop an approach for inferring the leaf angle distribution functions for corn canopy at four phenological stages, and (2) retrieve corn canopy LAI using the inferred leaf angle distributions obtained from terrestrial LiDAR for these four phenological stages.

## 2. Materials and Methods

### 2.1. Study Area

The study area is located in Baoding City, Hebei Province, China, ranging from 39.08°N, 115.48°E to 39.58°N, 116.23°E (Figure 1a). The field is cultivated in a rotation with corn and winter wheat (*Triticum aestivum*); soybean (*Glycine max*), cotton (*Gossypium hirsutum*), and sweet potato (*Ipomoea batatas*) are grown occasionally. Corn is generally sown at the beginning of June and harvested in the middle of September, that is, a 3-month growing period. The soil is fertile and the summer is hot and rainy. Thus, the study area is the main planted area for corn in North China.

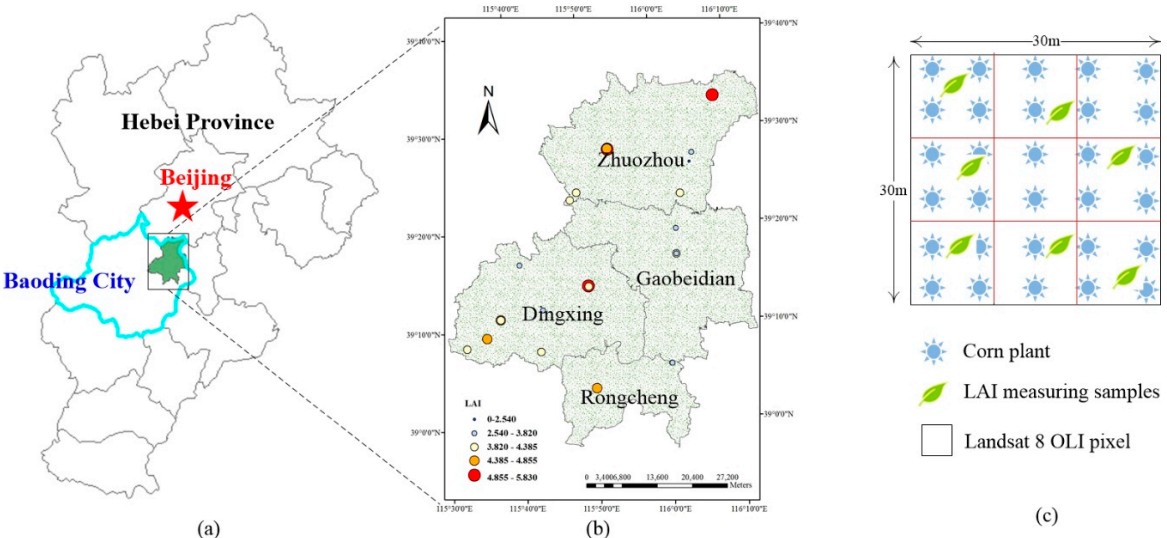

**Figure 1.** Location of the study area (**a**), the distribution of sampling quadrats in field work (**b**), and sampling pattern used in the fieldwork (**c**).

### 2.2. Data Collection

We measure the corn canopy LAI, chlorophyll content, leaf angles, leaf reflectance, canopy reflectance and background soil reflectance in sampling quadrats (Figure 1b). At each sampling site, we established eight study plots, each 30 m × 30 m in size (to match the resolution of Landsat pixels), with each plot divided into nine 10 m × 10 m quadrats for LAI measurement. Locations were determined using the scanner's onboard GPS receiver, with a 2D accuracy of 10m, and were used to register the study plots to the corresponding Landsat pixels.

#### 2.2.1. TLS Data Acquisition and Pre-Processing

Laser scanning was conducted with a Focus3D X330 scanner (FARO Technologies, Rugby, U.K., Figure 2a) at the same time as Landsat 7 ETM+ and Landsat 8 OLI image acquisition dates. The TLS

scanner employs a 1550-nm laser within a maximum field of view of 300° vertically by 360° horizontally (Table 1, Figure 3b). Its scanning speed is 122,000 to 976,000 points/sec, with a scanning distance of 0.6 m to 330 m. In each quadrat, plot-level TLS scans were performed from three positions outside each plot (Figure 3a). There is one scanning position located inside the plot. Four white balls were placed inside each plot and scanned as registration points; their positions were subsequently used to permit the co-registration of the point clouds generated from different positions because the data obtained from the different scan positions had different coordinate systems. Figure 2 provides photographs of the fieldwork.

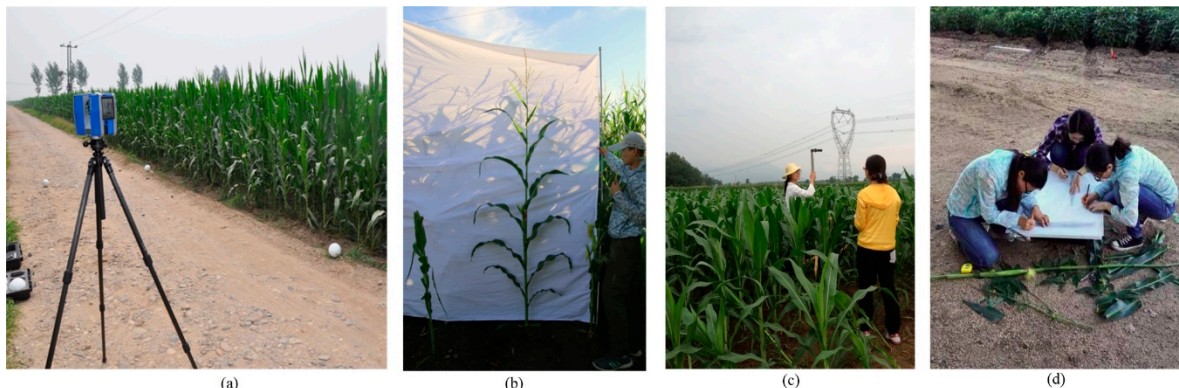

**Figure 2.** *In-situ* data collection activities conducted in each sampling plot: (**a**) obtaining a 3D image of the plot using the TLS scanner, (**b**) photographing the leaves for manual calculation of leaf angles, and (**c**,**d**) measuring LAI using the LI-COR LAI-2200C plant canopy analyzer.

**Table 1.** The parameters of the FARO Focus3D X330 LiDAR surveying instrument.

| Parameter | Range of Values |
|---|---|
| Scanning distance (m) | 0.6 to 330 |
| Scanning speed (points/s) | 122,000 to 976,000 |
| Ranging error (mm) | $\pm 2$ |
| Resolution (pixels) | $7 \times 107$ |
| Vertical field of view (°) | 300 |
| Horizontal field of view (°) | 360 |
| Laser class | Class 1 |
| Wavelength (nm) | 1550 |
| GPS | Integrated GPS receiver |

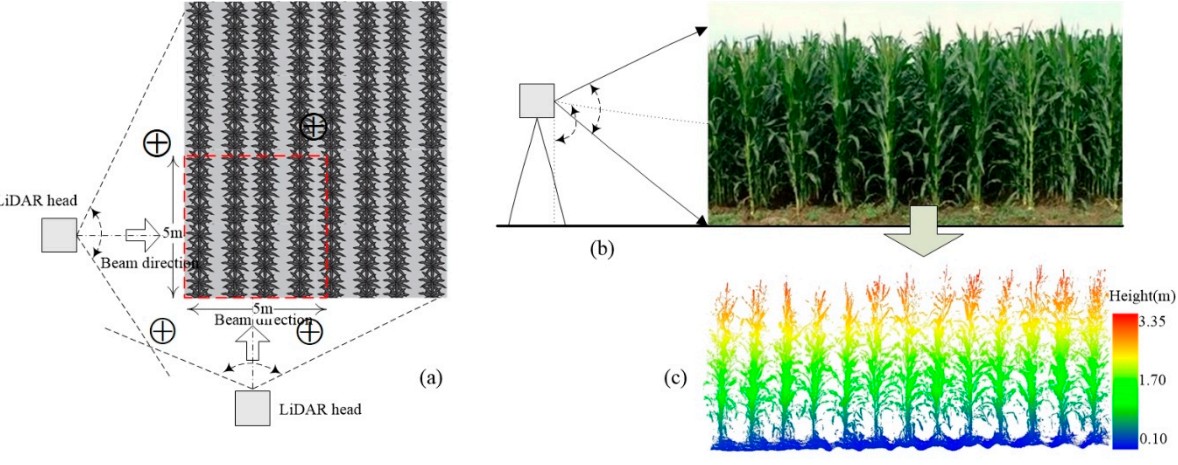

**Figure 3.** Top view (**a**) and side view (**b**) of the laser scanning scheme for the corn canopy, and (**c**) sample image of the scanning data for a corn canopy.

In pre-processing, data points representing "noise" associated with real returns from the corn canopy were removed. The noise points are mainly "flying points" for corn plants, because most leaf surfaces are inclined and its normal vectors are not parallel to the laser beam, which will result in the deviation between the received laser pulse and the emitted pulse. The points with this deviation greater than 30 will be classified as "flying points" [26]. The remaining points were co-registered and merged into a single composite point cloud per plot using the FARO SCENE software (www.faro.com). Figure 3c illustrates a sample result of the scanning.

To alleviate the contribution of non-photosynthetic materials to radiation interception, the difference of normals (DoN) method [27,28] is used to identify the points of corn leaves and stalk. Open-source code for the DoN operator was obtained from the Point Cloud Library (PCL; http://pointclouds. org) [29]. This identification is only done on field samplings and only for corn leaf angle distribution function computation.

### 2.2.2. Acquisition of Remote Sensing Images

Referring to corn's growing season, we collected two Landsat 7 ETM+ images on 10 July (the stem elongation stage) and 26 July (the heading stage), and two Landsat 8 OLI images on 19 August (the flowering stage) and 4 September (the grain-filling stage) in 2014. All images were downloaded from the United States Geological Survey gateway (https://www.usgs.gov/). The bands used in this study were blue, green, red, and near-infrared (NIR). Table 2 provides details of these four images, the sun elevation and azimuth angle, and the viewing zenith and azimuth angle, which are the inputs of PROSAIL model. We pre-processed the remote sensing images in the whole study area for geo-registration, radiometric calibration, and atmospheric correction to improve the accuracy of the LAI retrieval. The corn planted area is extracted using a decision tree and a mixed-pixel un-mixing classifier in our previous work [30]. The decision tree is built using the NDVI difference calculated from Landsat 8 OLI images on 19 August and 4 September. The pixels with $NDVI_{19\ August} > 0.6$ and $NDVI_{4\ September} > 0.55$ are classified as the corn planted area by testing the NDVI difference of the corn planted area and the non-corn planted area in the whole study area.

**Table 2.** The parameters of the Landsat 7 and 8 images.

| Date | Sensor | UTM Time | Sun Elevation Angle (°) | Sun Azimuth Angle (°) | Viewing Zenith Angle (°) | Viewing Azimuth Angle (°) |
|---|---|---|---|---|---|---|
| 10 July | ETM+ | 02:51:27 | 64.77 | 124.82 | 0 | 90 |
| 26 July | ETM+ | 02:51:31 | 62.52 | 128.55 | 0 | 90 |
| 19 August | OLI | 02:53:59 | 58.03 | 138.33 | 0 | 90 |
| 4 September | OLI | 02:54:02 | 53.72 | 144.98 | 0 | 90 |

### 2.2.3. Field Data Collection

In order to validate if the inferred leaf angle distributions can be used to improve LAI retrieval, an extensive field campaign was carried out on 4 September, 2014 to measure LAI using a LAI-2200 Plant Canopy Analyzer (LI-COR, Lincoln, NE, USA) (Figure 2c). We used the cover cap with a 45° field angle to eliminate the effect of non-plant objects within the range of the sensor's field of view. The skylight was measured one time and the light under leaves was measured four times, respectively, for every sampling corn plant. We obtained six measurement values in each quadrat, following a zigzag pattern (Figure 1c). The measured LAI is the effective LAI.

In addition to LAI, we also measured the leaf chlorophyll content using a SPAD-502 leaf chlorophyll meter on 4 September 2014. For this measurement, we used five corn plants and six leaves per plant to measure the chlorophyll content. For each leaf, we measured the chlorophyll content near the leaf tip, leaf bottom, and middle of the leaf, and used the average value to estimate the chlorophyll content of the leaf. The measured relative SPAD values are transformed to absolute

chlorophyll concentrations using the exponential fit model proposed by Markwell et al. [31]. The mean of all transformed chlorophyll values for each quadrat represented the actual chlorophyll content.

Finally, we measured the leaf angle by photographing the plants (Figure 2b) and manually analyzing the leaf angles in the images. The leaf reflectance, canopy reflectance and background soil reflectance are measured using SVC HR-1024 spectrometer (Spectra Vista Corporation, Poughkeepsie, NY, USA). The location of all the samplings was provided using a Huace i80 real-time kinematic (RTK) GPS receiver (Huace Ltd., Shanghai, China).

### 2.3. LAI Retrieval

We use the PROSAIL model to retrieve corn canopy LAI based on the remote sensing images acquired on 10 July, 26 July, 19 August and 4 September, 2014. We focus on testing if the introduction of the leaf angle distribution estimated from TLS data on each date will improve the accuracy of LAI retrieval in this study. Figure 4 summarizes the workflow of LAI retrieval in this study.

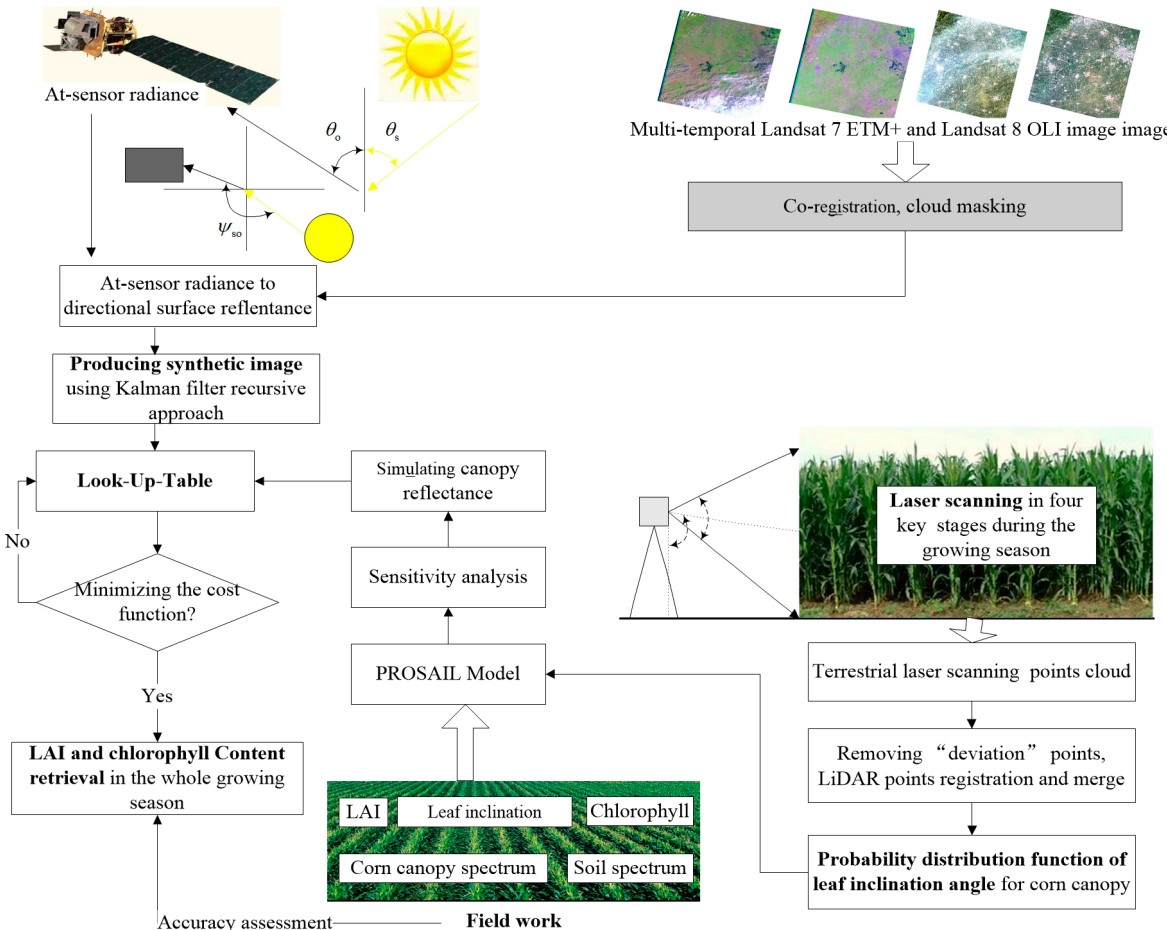

**Figure 4.** Overview of the LAI retrieval based on improving the leaf angle distribution function of the PROSAIL model.

### 2.3.1. PROSAIL Model and Sensitivity Analysis

PROSAIL is the coupling of the PROSPECT model and SAIL model, which simulates the bidirectional reflectance at the top of the canopy in the spectral range from 400 nm to 2500 nm as a function of input variables that relate to the structure of the canopy, the leaf optical properties, the background soil reflectance, and the solar geometry [7,11]. It assumes that the canopy is a turbid

medium in which leaves are randomly distributed [32]. The forward simulation expression of PROSAIL model is as followed:

$$\rho = \text{PROSAIL } (N, C_{ab}, C_{ar}, C_w, C_m, \text{LIDFa, LAI, hspot, tts, tto, psi, } \rho_{soil}) \tag{1}$$

From this formula, we can see that there are four kinds of input variables for PROSAIL model: leaf optical properties, canopy structure, background soil reflectance and sun-view geometry (see Table 3). Leaf optical properties is described by the mesophyll structural parameter (N), leaf chlorophyll content ($C_{ab}$), leaf carotenoid content ($C_{ab}$), dry matter ($C_m$), and equivalent water thickness ($C_w$). Leaf chlorophyll ($C_{ab}$) is measured with a SPAD-502 chlorophyll meter in this study. Equivalent water thickness ($C_w$) is tied to the difference of fresh leaf weight and dry leaf weight ($C_w = (C_{fresh\ leaf} - C_{dry\ leaf})/LAI$). Canopy structure is characterized by LAI, leaf angle distribution function (LIDFa) and the hot-spot parameter (hspot). LAI comes from the measured LAI value in field work using LAI-2200 Plant Canopy Analyzer. Leaf angle distribution function (LIDFa) comes from the calculated function using TLS points data of corn canopy in this study. The effect background soil ($\rho_{soil}$) is descripted by the soil reflectance. The sun-view geometry is described by the solar zenith angle (tts), view zenith angle (tto), and the relative azimuth angle between sun and satellite sensor (psi).

In order to constrain the behavior of the PROSAIL model and mitigate the ill-posed problem of LAI retrieval [33], the sensitivity analysis of the model is done firstly during LAI retrieval process. In other words, the sensitivity analysis is done to evaluate the response of canopy reflectance at any wavelength by changing input variables of PROSAIL model. Sensitive input variables can strongly affect model outputs, whereas, insensitive variables have much less impact on the results. The sensitivity analysis is done by running the PROSAIL model using a given range of one input variables within a certain interval (see Table 3), and other input variables are set as unique values. For an example, the PROSAIL model will be run using a range value [0, 8] with 0.01 interval for LAI, and other inputs such as LIDFa, hspot, N, $C_{ab}$, $C_{ar}$ etc. are set as default values when we analyze the sensitivity of LAI input variable. If the simulated reflectance is obviously different using a range of LAI and default values of other variables (i.e., LIDFa, hspot, N, $C_{ab}$, $C_{ar}$ etc.), we can conclude that LAI is a sensitive input parameter.

**Table 3.** Range and distribution of the input variables used to establish the synthetic corn canopy reflectance database in the lookup table.

| | | Model Variables | Range or Value | Distribution |
|---|---|---|---|---|
| | LAI | Leaf area index ($m^2\ m^{-2}$) | 0.1 to 7.0 | Uniform |
| Canopy | LIDFa | Leaf angle distribution (°) | 0 to 90 | Gaussian |
| | hspot | Hotspot parameter ($m\ m^{-1}$) | 0.1 | - |
| | N | Leaf structural parameter in PROSPECT | 1.518 | - |
| | $C_{ab}$ | Chlorophyll a+b content in PROSPECT ($\mu g\ cm^{-2}$) | 0.1 to 60.0 | Uniform |
| Leaf | $C_{ar}$ | Carotenoid content in PROSPECT ($\mu g\ cm^{-2}$) | 8 | - |
| | $C_w$ | Equivalent water thickness in PROSPECT (cm) | 0.05 to 0.3 | Gaussian |
| | $C_m$ | Dry matter content in PROSPECT ($g\ cm^{-2}$) | 0.002 to 0.012 | Gaussian |
| Soil and sky | $\rho_{soil}$ | Soil reflectance assumed to be Lambertian (1) or not (0) | 0–1 | Gaussian |
| | skyl | Ratio of diffuse to total incident radiation | Calculated by tts | - |
| | tts | Solar zenith angle (°) | / | / |
| Sun-sensor | tto | Viewing zenith angle (°) | / | / |
| | psi | Relative azimuth angle (v) | / | / |

The symbol "/" represents values obtained from the Landsat data header, "-" represents the one set value, and skyl is calculated by tts using $skyl = 0.847 - 1.61 \times \sin((90 - tts) \times rd) + 1.04 \times \sin((90 - tts) \times rd) \times \sin((90 - tts) \times rd)$.
**Notes**: The boundary conditions for the corn canopy, leaves, and soil were selected to describe the characteristics of all growth conditions for the corn canopy in our study area. Sun and sensor viewing conditions corresponded to the measurement conditions when the satellite passed overhead. All possible combinations of all variables were calculated from the corn leaf and canopy input ranges.

### 2.3.2. Leaf Angle Distribution Function Inferred from the TLS Data

An accurate leaf angle distribution function is needed to calculate the radiation flux densities at the leaf surfaces [16,34]. Thus, we focused on improving only the inclination angle density function in this study. One of the most popular mathematical angular distribution models was developed by Campbell [16], who approximated the angular distributions by adjusting the ratio of the semi-long axis (*b*) to the semi-short axis (a) of a geometrical ellipsoid. The ellipsoidal distribution of leaf angles is represented as follows:

$$g(\alpha) = \frac{2\chi^3 \sin\alpha}{\Lambda (\cos^2\alpha + \chi^2 \sin^2\alpha)^2} \tag{2}$$

where $\alpha$ is the leaf inclination angle ($0 \leq \alpha \leq \pi/2$), $\chi = b/a$, and

$$\Lambda = \begin{cases} \chi + \left(\sin^{-1}\varepsilon\right)/\varepsilon & \chi < 1 \\ \chi + \frac{\ln\left[\frac{(1+\varepsilon)}{(1-\varepsilon)}\right]}{2\varepsilon\chi} & \chi > 1 \\ 2 & \chi = 1 \end{cases} \tag{3}$$

where $\varepsilon = (1 - \chi^2)^{\frac{1}{2}}$. Zheng and Moskal [19] found that the $\chi$ value could be calculated from the plant leaf angle distribution function inferred from discrete terrestrial LiDAR points as defined in Equation (3)

$$\chi^2 = \frac{1}{3\sin^2\alpha} + 1 \tag{4}$$

where $\alpha$ is the leaf inclination angle with the highest frequency. Using the TLS point cloud from each of the four scanning dates, we computed the specific $\chi$ value for each date by inferring the specific $\alpha$ computed from the normal operator in PCL. Keeping the stalk points will have an effect on the computation of leaf angle distribution functions. So we remove the stalk points from leaf points using DoN method for $\chi$ value calculation. The DoN operator is the difference of leaf points normal and stalk points normal. The corn stalk is erect from ground commonly, so the stalk points normal are horizontal. In comparison, the corn leaved will distributed in all directions and their normal are irregular. Therefore, there are distinctly different normal between the normal of corn leaves and stalk. The DoN operator is the difference of leaf points normal and stalk points normal [28], which is used to remove the stalk points from leaf points in this study. Then the computation of corn leaf angle distribution function using corn leaf points is as followed.

Step 1: Computation of centroid for points set from neighboring $p_i$.

For a given TLS point $p_i$, there are k neighboring points which can organized as a point set $P_i$ ($P_i = p_{i1}, p_{i2}, p_{i3}, p_{i4}, \ldots, p_{ik}$). A surface $T_i$ can be constructed using the points set $P_i$. $\vec{n_i}$ is the normal vector of surface $T_i$, and $\overline{p}_i$ is the centroid of points set $P_i$. And $\overline{p}_i$ is computed as:

$$p_i = \frac{1}{k} \sum_{i=1}^{k} p_{ik} \tag{5}$$

Step 2: Computation of normal vector for point $p_i$ using PCA method.

Firstly, the covariance matrix *M* of point set $P_i$ ($P_i = p_{i1}, p_{i2}, p_{i3}, p_{i4}, \ldots, p_{ik}$) is computed using PCA algorithm. Secondly, the eigenvalues ($\lambda_1, \lambda_2, \lambda_3$) and eigenvectors ($\vec{e_1}, \vec{e_2}, \vec{e_3}$) of covariance matrix *M* are calculated. Lastly, exploring the eigenvectors of the critical eigenvalues, which is the normal vector for point $p_i$. The covariance matrix *M* is computed as:

$$M = \frac{1}{k} \sum_{i=1}^{k} (p_i - \vec{p})(p_i - \vec{p})^T \tag{6}$$

where, $(p_i - \bar{p})$ is a column vector; $(p_i - \bar{p})^T$ is the transpose of column vector; $\bar{p}$ is the centroid of neighboring points.

Step 3: Eliminating the directional ambiguity of normal vectors.

There is directional ambiguity randomly for all points: there are two equal but opposite (negative and positive) normals for any object surface, both of them are mathematically valid. Disambiguation of the normals is done by constructing the Riemann surface, and the normal with the minimum spanning tree in the Riemann surface is the direction of the disambiguated normal for the given point.

Step 4: Computation of leaf angle distribution function.

Normal vectors for each point cloud are calculated by step 1 to step 3. Therefore, the corn leaf angle distribution function can be calculated by counting all leaf points normal, and the leaf angle is ranging from 0° to 90°.

### 2.3.3. LUT-Based LAI Retrieval Strategy Based on PROSAIL Model

1. LUT generation

LAI retrieval using PROSAIL requires adjusting the values of the input canopy biophysical variables to best match the bidirectional reflectance factors measured by the sensor for a range of directions and wavelength bands [35]. However, retrieving biophysical parameters from PROSAIL is nontrivial. First, the model uses a nonlinear function of the canopy optical and structural parameters. Second, the retrieval solution is not always unique because two or more combinations of canopy parameters may produce similar canopy reflectance spectra [36]. Furthermore, there are uncertainties in the sensor measurements and the input biophysical variables of the canopy model [27]. Therefore, LAI retrieval using PROSAIL is "an ill-posed problem" [35,37,38]. To solve this problem, several algorithms based on minimization have been developed [39], including lookup tables, quasi-Newton algorithms, neural networks, and tangent linear models. Because of it's ease of use and general robustness, we used the LUT approach in this study.

The determination of LUT dimension is an important issue for LUT generation for LAI retrieval [40]. In order to take balance between the flexibility in determining the optimal band combination for LAI retrieval and mitigation of ill-posed problem, the LUT dimension is designed ranging from blue band to NIR band. Consequently, the spectral wavelength ranging from 450 nm–900 nm is simulated by the PROSAIL model and used to generate LUT, which produces the size of the LUT in column direction. Subsequently, the size of the LUT in row direction should be specified, which is determined by the range/value and interval of input variables [40]. The sun-view geometry (tts, tto, psi) and skyl are coming from the head files of remote images, which are unique values. So, there are only three kinds of parameters in our lookup table: leaf optical properties, canopy structure features and $p_{soil}$. Weiss et al. [41] found that an LUT based on 100,000 modeled spectra provides an optimal compromise between model accuracy and required computer-resources. Therefore, we combine the input variables using the following range/value and interval of input variables (see Table 3). The version of PROSPECT model used in this study is PROSPECT-5B [42], so the brown pigments ($C_{bp}$) is fitted to 0 [43], which is not listed in Table 3.

2. Cost function

Based on the generated LUT, the program identifies the remote sensing image pixel by pixel. According to the range/value of input variables (see Table 3), there are about 100,000 spectra loaded from the LUT library for LAI retrieval. We compare the pixel values from the blue band to the near infrared band of the Landsat images with the spectrum ranging from 450 nm–900 nm in the lookup table, aiming at finding the best fit(s) for LAI retrieval. The cost function is used to measure the discrepancies between the observed pixel values and the simulated reflectance values in the lookup table. The pair with minimum discrepancy is the best fit/match, which will be stored as the pixel value of LAI output image [40]. Referring to previous studies [35,40], the cost functions based on the

root-mean-square error (RMSE) between simulated and observed reflectance values at multiple bands were used for LAI retrieving in this study, which is defined as:

$$\text{RMSE} = \sqrt{\frac{1}{n} \sum_{\lambda=1}^{n} \left( R_{\text{sim}}(\lambda) - R_{\text{L}}(\lambda) \right)^2} \tag{7}$$

where $R_{\text{L}}(\lambda)$ is the reflectance in band $\lambda$ for the Landsat image, $R_{\text{sim}}(\lambda)$ is the simulated reflectance in band $\lambda$, and n is the number of wavelength bands. The retrieved LAI is found when the RMSE approaches 0.

## 3. Results and Analysis

### 3.1. Sensitivity Analysis of PROSAIL for Simulating Corn Canopy Reflectance

The sensitivity analysis is done to determine which inputs are sensitive parameters of PROSAIL model and if the leaf angle distribution function (*LIDFa*) is a sensitive variable [11]. And the sensitive parameters would be set as a ranging value for LUT construction. Figure 5 is the sensitivity analysis result for the 12 input parameters of the PROSAIL model in the blue, green, red and NIR bands. Figure 5a–c,k,l show that the simulated reflectance differs obviously when LAI, *LIDFa*, N, *tts*, and *tto* are changed and other inputs are fixed. This shows that LAI, *LIDFa*, *tts*, and *tto* are sensitive input parameters within blue, green, red and NIR bands. Figure 5f–h show that the simulated reflectance differs within NIR band when $C_w$, $C_m$ and hspot are changed and other inputs are fixed. This shows that $C_w$, $C_m$ and hspot are sensitive within NIR band. Figure 5e,i,j show that the simulated reflectance is few different within all bands. This shows that $C_{ar}$, psoil and skyl are insensitive inputs for the PROSAIL model.

### 3.2. Inferred Leaf Angle Distribution Function from the TLS Scanner Data

We obtained leaf angle distribution functions for the four phenological stages from the TLS scanner data. The χ values simulated by using the Campbell ellipsoid were 1.223 on 10 July, 1.206 on 26 July, 1.214 on 19 August, and 1.195 on 4 September (Table 4). Figure 6 shows the leaf angle distribution functions for the corn canopy modeled as an ellipsoid distribution $(b > a)$.

**Table 4.** Proportion of the leaf angles (in 10° intervals) on the four phenological dates.

| Date | χ | Proportion of Leaf Angles (% of Total) | | | | | | | | |
|---|---|---|---|---|---|---|---|---|---|---|
| | | 0°–10° | 10°–20° | 20°–30° | 30°–40° | 40°–50° | 50°–60° | 60°–70° | 70°–80° | 80°–90° |
| 10 July | 1.223 | 7.83 | 11.33 | 14.2 | 16.1 | **16.68** | 15.05 | 10.63 | 5.68 | 2.5 |
| 26 July | 1.206 | 6.49 | 8.98 | 10.73 | 12.42 | 14.18 | **15.41** | 14.67 | 11.1 | 6.02 |
| 19 August | 1.214 | 7.1 | 10.11 | 11.3 | 11.89 | 13.4 | **14.79** | 14.02 | 10.85 | 6.54 |
| 4 September | 1.195 | 6.17 | 8.76 | 10.02 | 10.73 | 12.19 | 14.54 | **15.79** | 13.53 | 8.27 |

Notes: Boldfaced values represent the interval with the highest proportion of the total.

Figure 6 indicates that the leaf angle at the maximum probability density decreased when χ increased through time: the leaf angle intervals around the maximum probability density are from 40° to 50° on 10 July (the stem elongation stage), from 50° to 60° on 26 July (the heading stage) and on 19 August (the flowering stage), and from 60° to 70° on 4 September (the grain-filling stage). Thus, the dominant leaf angle increased over the course of the growing season.

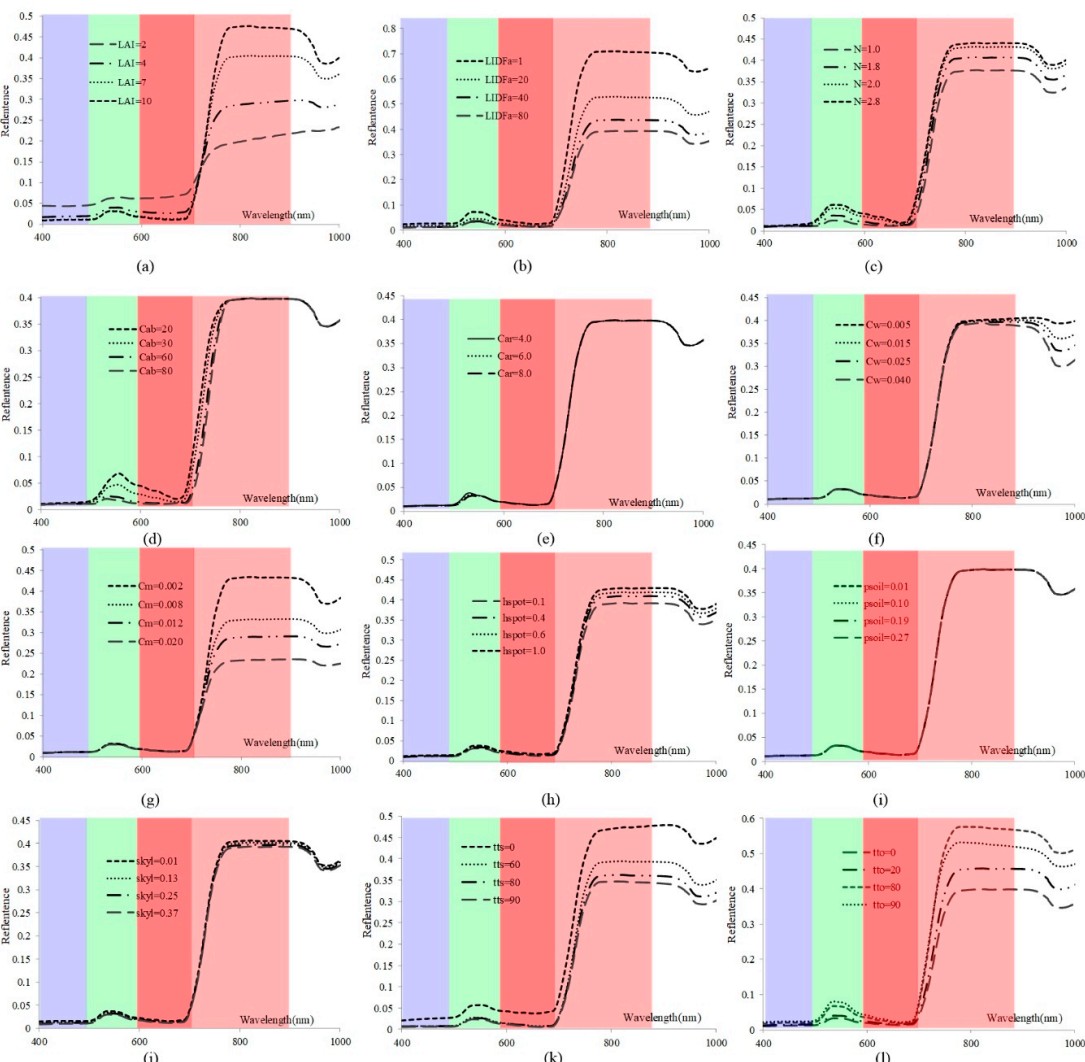

**Figure 5.** Results of the sensitivity analysis for the 12 input parameters to which PROSAIL is sensitive.

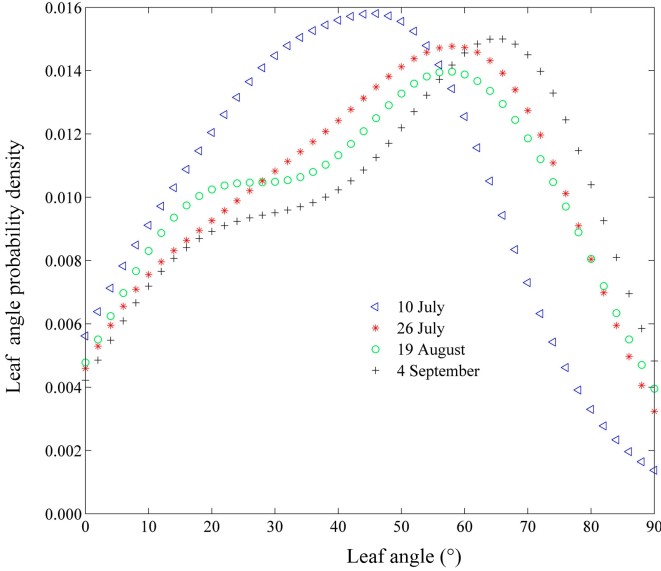

**Figure 6.** The leaf angle probability density functions for the corn canopy on 10 July, 26 July, 19 August, and 4 September, 2014, respectively.

### 3.3. Retrieved Corn Canopy LAIs

LAI values at four phenological stages were retrieved through using the PROSAIL model based on the lookup table and the leaf angle function inferred from the terrestrial LiDAR data. Figure 7 shows maps of the retrieved LAI derived for the stem elongation stage (10 July), the heading stage (26 July), the flowering stage (19 August), and the grain-ripening stage (4 September). Figure 7a shows low LAI values during the early growing season (10 July, at the seven-leaf stage for corn in study area), with values ranging from 0.5 m$^2$/m$^2$ to 2.7 m$^2$/m$^2$. By 26 July (heading stage; Figure 7b), LAI had increased to 3.7. By late August (flowering stage; Figure 7c), LAI reached values of up to 7.2. By early September (grain filling stage; Figure 7d), LAI reached its seasonal maximum, with values reaching up to 7.8. The retrieved LAI maps therefore show a realistic progression of LAI throughout the growing season.

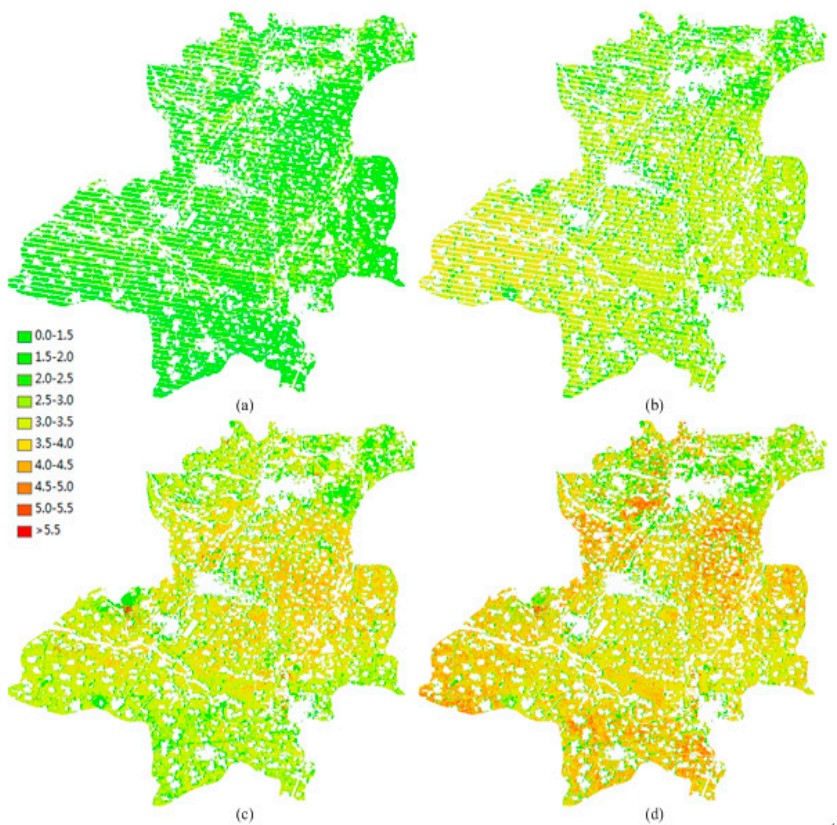

**Figure 7.** The LAI maps derived using the inferred leaf angle function from TLS scanner data on (**a**) 10 July (stem elongation stage), (**b**) 26 July (heading stage), (**c**) 19 August (flowering state), and (**d**) 4 September (grain-filling stage).

In order to validate these LAI retrieval results, a further assessment of the accuracy is completed. The extensive in-situ measured LAIs on 4 September are used to determine whether the introduction of inferred leaf angle distributions from TLS data points can improve the LAI retrieval accuracy using the PROSAIL model. In addition, the LAI retrieval results on 10 July, 26 July, 19 August and 4 September with default and inferred corn leaf angle distribution functions are all compared with MODIS LAI products in the whole study area. This comparison is done to evaluate the LAI retrieved results in a wide spatial range and temporal range. Figure 8 shows the relationships between the in-situ measured LAI and the retrieved LAIs using inferred and default leaf angle function for the retrieved results of 4 September. The results show a strong and significant fit for the retrieved LAI using leaf angle function inferred from TLS scanner data ($R^2 = 0.82$, $p < 0.001$) and the retrieved LAI using default Campbell leaf angle function ($R^2 = 0.76$, $p < 0.001$). Both retrieved values did not differ significantly from in-situ

values (*t*-test, $p < 0.001$). Moreover, we found a low RMSE value of 0.31 m$^2$/m$^2$ for the retrieval using leaf angle function inferred from TLS scanner data, compared with the RMSE value of 0.56 m$^2$/m$^2$ for the retrieval using the default Campbell leaf angle function. In addition, the difference between the retrieved LAI using inferred leaf angle function and in-situ value ranged from 0 to 0.6 m$^2$/m$^2$, and the difference between the retrieved using default Campbell leaf angle function and in-situ value ranged from 0 to 1.0 m$^2$/m$^2$. This comparation revealed that the use of leaf angle distribution inferred from TLS data worked better than the default value for this retrieval.

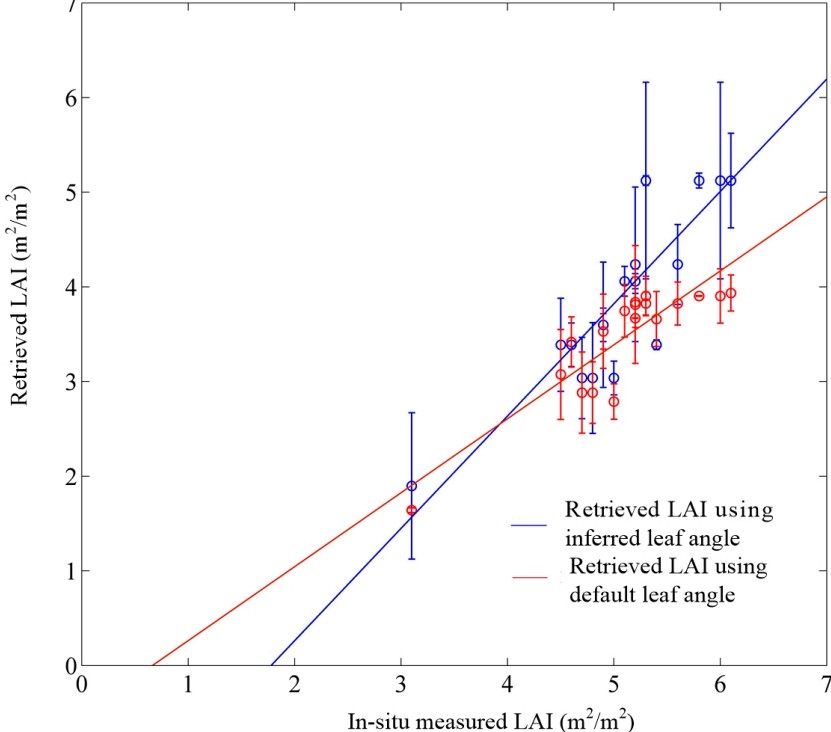

**Figure 8.** The relationships between the in-situ measured LAI and the retrieved LAIs using inferred and default leaf angle function. Values represent means $\pm$ SD based on the in-situ LAI measurements at each sample plot.

To further evaluate the retrieved LAI results, we compared the LAI retrieving values with inferred and default leaf angle distribution function, Figure 9a is the difference between them for LAI retrieving results on 4 September 2014, i.e., LAI$_{\text{with inferred LAD}}$-LAI$_{\text{with default LAD}}$. Figure 9b is the zoomed difference LAI map in the middle of the study area. We can see that the LAI result with inferred leaf angle distribution using TLS data is higher than the LAI result with default leaf angle function. The difference between LAI$_{\text{with inferred LAD}}$ and LAI$_{\text{with default LAD}}$ is ranging from 0 m$^2$/m$^2$ to 1.5 m$^2$/m$^2$. This analysis shows that the use of leaf angle distribution inferred from TLS data increases the LAI retrieved result.

Finally, all LAI retrieved results using default leaf angle and referred leaf angle in the whole study area are compared with homologous MODIS LAI products. Figure 10 shows the compared results of LAI retrieved on 10 July, 26 July, 19 August, 4 September 2014, respectively. Blue boxes represent the retrieved LAI using default corn leaf angle, red boxes represent the retrieved LAI using inferred leaf angle distribution functions from TLS points, and green boxes represent the LAI value from MODIS LAI products. In both panels, solid lines within the boxes represent the mean, box top and bottom represent the 75[th] and 25[th] percentile, respectively. The comparison revealed that the LAI variation trend from 10 July to 4 September using inferred corn leaf angle distribution functions is more consistent with the MODIS LAI variation trend than the LAI variation trend using default leaf angle function. Both plots raised quickly from 10 July to 26 July when the corn grew quickly, and

both plots raised slowly from 26 July to 4 September when the corn growth was slow. In addition, the curve of retrieved LAI using inferred corn leaf angle distribution functions is higher than the MODIS LAI curve and the curve of retrieved LAI using default leaf angle. Previous comparisons proposed that the MODIS LAI products underestimate the LAIs of corn crops [44] and other vegetation species [10,45–48]. Fortunately, our experiment shows that this underestimate can be alleviated for corn canopy LAI retrieval by considering the leaf angle difference in different phenological stages.

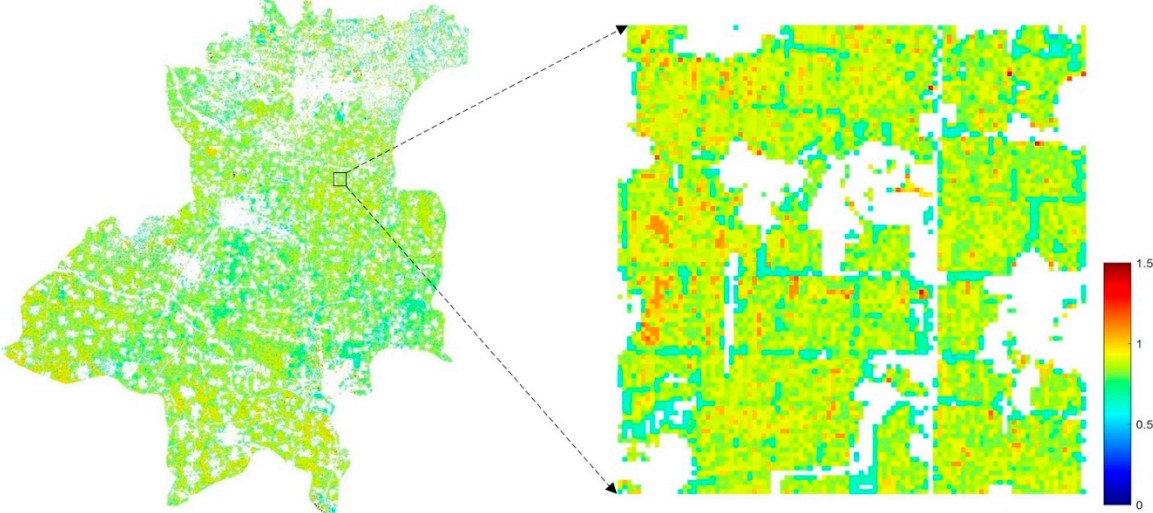

**Figure 9.** LAI bias ($m^2/m^2$) between the retrieved LAI with and without inferred leaf angle distribution function on 4 September 2014.

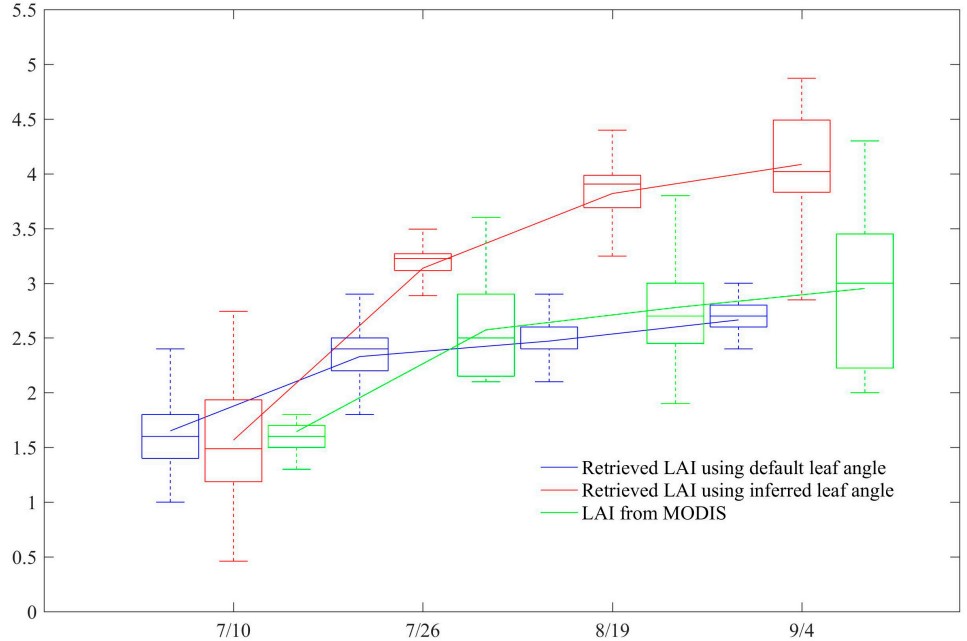

**Figure 10.** Comparation between the retrieved LAIs with default and inferred leaf angle distribution function on 10 July, 26 July, 19 August, 4 September, 2014, respectively.

## 4. Discussion

Previous research has documented the importance of leaf angle distribution function for LAI retrieval using the PROSAIL model [11,16,18,34]. However, only the Campbell ellipsoidal leaf angle distribution function [18] is commonly used to characterize vegetation leaf angle by defining the ratio

of ellipse a and b of a fitted ellipsoid without considering the difference of leaf inclination between different vegetation species and different growing stage, as it is known that most vegetation leaf angle distributions are not precisely ellipsoid. In addition, the leaf inclination angles change during the whole growth period for some crops such as corn. This study proposes the method of estimating the leaf angle distribution functions from terrestrial LiDAR-scanned points at four phenological stages of corn for LAI retrieval using PROSAIL model. Millimeter-scale measurement precision of terrestrial LiDAR points ensures the accuracy of leaf angle distribution function.

Sensitivity analysis results reveal that the *LIDFa* is a sensitive input from blue band to NIR band for the PROSAIL model. These four bands are the commonly used bands for LAI retrieval. This indicates that the description of the leaf angle distribution function will strongly affect the LAI retrieving accuracy. Therefore, the attempt of inferring leaf angle distribution function from the terrestrial laser scanner points data in this study is meaningful for LAI retrieval based on the PROSAIL model. At the same time, the specification of leaf inclination angle does good to mitigate the ill-posed problem of LAI retrieval [49].

This study has demonstrated that the leaf angle distribution functions are different for the corn plants at different phenological growing stages. Our results show that the maximum corn leaf angle is ranging from 40° to 50° on 10 July (stem elongation stage). The uppermost leaf is almost up-right and the other 6–8 leaves are oblique upward. During the growing period, the corn leaves will flat gradually for intercepting more sunlight. Thus, the maximum corn leaf angle changes from 50° to 60° on 26 July (heading stage) and 19 August (flowering stage) in this study. In the coming growing season, the corn leaf will be curled and the leaf tip will hang down gradually. And the maximum corn leaf angle changes from 60° to 70° on 4 September (grain-filling stage) for our studied corn plants. This leaf angle distribution function changing is in line with the corn leaf growing rhythm. This result confirms the assumption that the leaf angle distribution of corn canopy is clearly changing during different phenological phases over the growing season.

Our LAI retrieved results in four corn phenological stages show that the incorporation of derived specific corn leaf angle distribution functions distributes the improvement of LAI retrieval using the PROSAIL model. Two accuracy assessments are completed. One is the comparison of the retrieved LAIs using inferred and default leaf angle distribution functions with the measured LAIs on the in-situ measuring quadrats on 4 September, 2014. The comparison revealed that the accuracy of LAI retrieval using the inferred leaf angle function from TLS scanner data is higher than using the default Campbell leaf angle function. This improvement is a result of the fact our method takes into account the phenological change of corn leaf angle distributions using TLS points echoing on corn leaves. Of course, there are relevant input variables are dependent on the phenological stage and influence LAI retrieval next to leaf angle, such as leaf carotenoid content ($C_{ab}$), dry matter ($C_m$) etc. We are focusing on depicting the leaf angle change in this study. And we will expand the LAI retrieval improvement by bringing in other input variables in future work. The other validation is done in a wide spatial range (the whole study area) and multi-temporal (10 July, 26 July, 19 August and 4 September, 2014). Comparison results show that the spatial pattern and temporal trajectory of LAI retrieved results using inferred corn leaf angle distributions are closer to MODIS LAI products. This conclusion reveals that the LAI retrieval improvements can exist in a time series LAI retrieval and generalize to wider corn planted areas. Much research has indicated that the retrieved LAI using a remote sensing technique is underestimated [50–54], especially for MODIS LAI [10,44–48]. Therefore, the inferred leaf angle distribution improves the underestimation of LAI retrieval using remote sensing images.

This study demonstrates the ability of derived corn leaf angle distribution function to improve LAI retrieval accuracy using PROSAL model. This method leads to the improvement of LAI retrieval using a physical model. However, some limitations are worth noting. One is that we only depict the leaf angle distribution function for corn canopy, and there is no other crop studied currently. We will extend this method to other crops, such as sugarcane, wheat, rice, potato etc. The other limitation is that the terrestrial LiDAR scanner is costly. Namely, the LiDAR observations are valuable and often

limited in availability. Therefore, we will explore the possibility of deriving the leaf angle distribution functions of different phenological dates by taking stereoscopic photographs using a photographic measurement method in future work.

## 5. Conclusions

In this study, we propose a method for retrieving multitemporal corn canopy LAI values by using a TLS scanner data to infer an empirical leaf angle probability density function suitable for use in the PROSAIL model. The novelty of this study is that different empirical leaf angle distribution functions for the corn canopy were estimated from terrestrial LiDAR data at four phenological stages. The results show that the inferred leaf angle function improved the accuracy of LAI retrieval compared with the default Campbell leaf angle function. In addition, we localized the PROSAIL model to retrieve corn canopy LAI by accounting for its sensitivity to changes in parameters such as the leaf angle distribution function, LAI, and leaf chlorophyll (a+b) content.

The retrieved LAI results demonstrate the potential of retrieving multitemporal corn canopy LAI by inferring specific leaf angle distribution functions for use in the PROSAIL model and the ability of this method to account for phenological changes during the growing season. By adopting this method, it should be possible to improve the monitoring of crop growth, as well as the detection of stresses such as drought and yield prediction. Future studies can be focus on the following directions. First, exploring how to infer other sensitive inputs to improve vegetation canopy parameter retrieval, and ultimately resolve the ill-posed problem. Second, harmonizing the prior knowledge of crop growth to the canopy parameter retrieval, for example, LAI change, chlorophyll content change etc. in the whole growing season. (3) Developing different cost functions to account for model effects, including 4D-VAR method, which considers observation errors and model errors to determine the best fits between observed and simulated reflectance signatures.

**Author Contributions:** This work is cooperated by our research team, and the contributions are as followed. Conceptualization, W.S. and J.H.; methodology, W.S. and M.Z.; software, M.Z.; writing—original draft preparation, W.S.; writing—review and editing, D.L.

**Funding:** This study was funded by the National Natural Science Foundation of China under the project 'Growth process monitoring of corn by combining time series spectral remote sensing images and terrestrial laser scanning data' (41671433), 'Estimating the leaf area index of corn in whole growth period using terrestrial LiDAR data' (41371327), Science and Technology Facilities Council of UK- Newton Agritech Programme (Project No. ST/N006798/1) and the National Keypoint Research and Invention Program of the Thirteenth 5-year Plan (2017YFD0300903).

**Conflicts of Interest:** The authors declare no conflict of interest.

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
