# Peer review of "Retrieving Corn Canopy Leaf Area Index from Multitemporal Landsat Imagery and Terrestrial LiDAR Data"

_remotesensing, doi:10.3390/rs11050572_

Reviewer 1 Report

The Authors’ demonstrated  PROSAIL inversion based on probability distribution function of leaf inclination angle computed from TLS improve the accuracy of  LAI retrieval of corn canopy from remote sensing data (Landsat-8). However, the way the methods are written is a bit difficult to comprehend. I suggest modifying the way the methodology is presented as well as the abstract. They have to clearly show how the PROSAIL model was inverted with the two LADF and present the results from the two approaches side by side starting from the abstract.

Line 20: replace “distributes to” by ‘contribute to’ or similar words.

Line 21 – 25: It seems the study is a comparison between LAI retrieval based on the default leaf angle distribution function (LADF) and modified LADF from LiDAR. But, the presented result is only for the modified LADF.  where is the result obtained using the default leaf angle distribution function? How do readers understand this?

Table 3: please note leaf structural parameter (N), and leaf carotenoid content (Car) are not a parameter that can be obtained from satellite images header file. Remove the symbol “/” from those rows.

Line235 -244: very sloppy and difficult to understand. Rewrite it

Author Response

Dear Reviewer:

Thank you very much for your very helpful suggestions. And thank you for your patience waiting for the revised version of our manuscript. We have worked hard for this revision. And the followed are the specific responses to each comment, we hope that our responses and the resulting changes in the manuscript will be OK. However, we will be happy to resolve any remaining issues.

Sincerely,

Wei Su, Jianxi Huang, Desheng Liu, Mingzheng Zhang

1. the way the methods are written is a bit difficult to comprehend. I suggest modifying the way the methodology is presented as well as the abstract. They have to clearly show how the PROSAIL model was inverted with the two LADF and present the results from the two approaches side by side starting from the abstract.

Reply: we have rewritten the whole method part now, which has been labeled using red color from Line 188 to Line 322.

2. Line 20: replace “distributes to” by ‘contribute to’ or similar words.

Reply: the “distributes to” has been replaced by ‘contribute to’ now.

3. Line 21 – 25: It seems the study is a comparison between LAI retrieval based on the default leaf angle distribution function (LADF) and modified LADF from LiDAR. But, the presented result is only for the modified LADF.  where is the result obtained using the default leaf angle distribution function? How do readers understand this?

Reply: the LAI retrieving result using the default leaf angle distribution function has added now from Line 24 to Line 30.

4. Table 3: please note leaf structural parameter (N), and leaf carotenoid content (Car) are not a parameter that can be obtained from satellite images header file. Remove the symbol “/” from those rows.

Reply: yes, they are the given numbers, not same to the tts, tto, psi from Landsat data header file. So we have deleted these symbol “/” from rows of leaf structural parameter (N), and leaf carotenoid content (Car), hspot and skyl. Now, symbol “/” represents values obtained from the Landsat data header, and “-” represents the one set value. In addition , the skyl is calculated by solar zenith angle (tts) and we have corrected this too.

5. Line235 -244: very sloppy and difficult to understand. Rewrite it

Reply: this part has been rewritten now. The rewritten sentences are as followed: “The sensitivity analysis is done by running the PROSAIL model using a given range of one input parameter within a certain interval (see the list in Table 3) and fixing other input parameters. For an example, the PROSAIL model will be run using a range value [0, 8] with 0.01 interval for LAI and fixed default values for other inputs such as LAD, hspot, N,Cab, Car etc. when we analyze the sensitivity of LAI. If the simulated reflectance is obviously different using a range of LAI and fixed default values (i. e. LAD, hspot, N,Cab, Car etc.), we can conclude that LAI is a sensitive input parameter. Fig. 5 is the sensitivity analysis result for the 12 input parameters of PROSAIL model for the blue, green, red and NIR bands. Fig. 5 (a), (b), (c), (k) and (l) show that the simulated reflectance differs obviously when LAI, LIDFa, N, tts, and tto are changed and other inputs are fixed. This shows that LAI, LIDFa, tts, and tto are sensitive input parameters within blue, green, red and NIR bands. Fig. 5 (f), (g) and (h) show that the simulated reflectance differs within NIR band when Cw, Cm and hspot are changed and other inputs are fixed. This shows that Cw, Cm and hspot are sensitive within NIR band. Fig. 5 (e), (i) and (j) show that the simulated reflectance is few different within all bands. This shows that Car, psoil and skyl are insensitive inputs for PROSAIL model.”

Reviewer 2 Report

A relevant and interesting aspect on the relevance of the leaf angle distribution for LAI retrieval from RTMs is presented and elaborated in an experiment. This is certainly a topic which requires attention. However, in the current paper on several points in the introduction, description of methods and results, and especially also in the discussion more in depth but concise elaboration is required. Especially the connection with previous work on this topic is currently to limited. In addition in the results section, the outcome of one date is presented while the other dates are not shown. To bring the paper to an acceptable level additional improvements are required for which detailed suggestions are provided below.

L75: lidar is shortly introduced: would be relevant to provide some more background on how leaf angle distributions can be derived from point cloud datasets. What is current status on this especially for crops like corn.

L97-202: from the description its not clear which instrument you used to measure LAI

L108-109: how did you decide on the number of scan positions: where you able to cover all corn plants in sufficient detail?

L114-115: its not yet clear what kind of noise has been removed, can you be more specific and also which threshold value was used for this

L137-138: are you referring here to planted area within the field plots? Which decision tree are you referring to?

L147: how did you transform the relative SPAD values to absolute chlorophyll concentrations: did you use published relations or analyzed chlorophyll concentrations in the lab?

L166-167: next to leaf angle distribution, are there also relevant parameters which are dependent on the phenological stage and thus could influence LAI retrieval. If so, how did you take this into account.

In Figure 4 next to Landsat also Modis and Sentinel-2 are mentioned: these sources were not used in this study? Better to remove from figure

L189-191: from this description its not clear if you made a distinction between leaf and stalk points and if you removed the stalk points when calculating the leaf angle distribution. Keeping the stalk points will have an effect on the retrieval of the leaf angle distribution.

Table 3: the symbol / is used to indicate two issues: 1) landsat information; and 2) not relevant for example in case of one set value. Better to use two different symbols for this to avoid confusion. Which steps did you apply for the variable parameters to fill your LUT? What was the number of entries in your LUT?

L218-228: a few points are unclear in this paragraph: why do you only take into account the green and near-infrared band for the inversion. How do you combine the RMSE values for the two bands to make a selection for the best retrieval. Often model inversion approaches result in ill-posed solution: how did you deal with this for this study?

L238: probably you mean here insensitive instead of sensitive

L249-250: the values indicated in the text differ from the values indicated in table 4. As you have repeated field measurements. To which extent are these values statistically different?

Figure 6: titles and units for x and y-axis are missing, also statement of date is missing in the caption

L275: here you show the results for the September image: what about the other dates?

Figure 8: titles and units for x and y-axis are missing

Figure 9: date is missing in caption: in L299 a different range is mentioned than is indicated in the legend to the detailed map

L330: this result does not validate the improved LAI retrieval, rather is confirms the assumption that the leaf angle distribution is clearly changing during different phenological phases over the growing season

L333-335: the current results only show the improvement (although not significant) for the observations in September as presented in Figure 6: the evaluation of the other dates is currently not presented and thus should be added to the results section to support this statement

L339-343: this part is a repetition from the results section: rather I would expect here a more elaborate assessment and explanation why the adapted leaf angle distribution would result in a better retrieval of LAI. Secondly, comparison with previous studies are currently completely lacking and need to be added to synthesise the results of this study with previous results. And third, the paper does not provide an outlook on how this result should be developed further. Lidar observations seem to be valuable but are often limited in availability. Can we derive reference values for the LAD of different crops according to fixed protocol of lidar measurements covering different crops around te world? This kind of elaboration would greatly contribute to improving the general applicability of RTMs for crop trait retrieval. Again also refer to previous studies as this work is not a stand alone research.

Author Response

Dear Reviewer:

Thank you very much for your very helpful suggestions. And thank you for your patience waiting for the revised version of our manuscript. We have worked hard for this revision. And the followed are the specific responses to each comment, we hope that our responses and the resulting changes in the manuscript will be OK. However, we will be happy to resolve any remaining issues.

Sincerely,

Wei Su, Jianxi Huang, Desheng Liu, Mingzheng Zhang

1. L75: lidar is shortly introduced: would be relevant to provide some more background on how leaf angle distributions can be derived from point cloud datasets. What is current status on this especially for crops like corn.

Reply: there are two kinds of methods for deriving plants leaf angle distributions currently. And we added the descripts of these two kinds of methods as the background of leaf angle estimation using TLS, which is from Line 82 to Line 84.

In addition, the introduction about the estimation methods of Zheng and Moskal, Vicari et al., Bailey and Mahaffee, Xu et al. and Li et al. are added from Line 84 to Line 94.

2. L97-202: from the description its not clear which instrument you used to measure LAI

Reply: the LAI-2200 Plant Canopy Analyzer was used to measure LAI, which is depicted in Line 163 to Line 169. And the used cover cap and measure method have been depicted too, which is depicted in Line 156 to Line 160.

3. L108-109: how did you decide on the number of scan positions: where you able to cover all corn plants in sufficient detail?

Reply: there are three outside and one inside scan position for laser scanning in sum. We have added this explain from Line 124 to Line 125 now.

4. L114-115: its not yet clear what kind of noise has been removed, can you be more specific and also which threshold value was used for this.

Reply: for the corn plants, the noise points are mainly “flying points”, there are recorded deviation values for points. We classified the points with deviation value great than 30 as “flying points” in line with previous research. So we add this description from Line 130 to Line 134 now.

5. L137-138: are you referring here to planted area within the field plots? Which decision tree are you referring to?

Reply: this is the planted area of corn in the whole study area, not only the field plots. So we polish this sentence now. The decision tree is built using the NDVI difference calculated from Landsat 8 OLI images on 19 August and 4 September. This is our previous work about corn planted area extraction, which has been added from Line 156 to Line 160 now.

6. L147: how did you transform the relative SPAD values to absolute chlorophyll concentrations: did you use published relations or analyzed chlorophyll concentrations in the lab?

Reply: we transformed the relative SPAD values are transformed to absolute chlorophyll concentrations using the exponential fit model proposed by Markwell et al., which has been added this description from Line 174 to Line 176 now.

7. L166-167: next to leaf angle distribution, are there also relevant parameters which are dependent on the phenological stage and thus could influence LAI retrieval. If so, how did you take this into account.

Reply: yes. Next to leaf angle distribution, other inputs such as LAI, Cm etc. change with phenological stage and influence LAI retrieval. What we want to describe is there are four remote sensing images acquired on different dates are used to retrieve LAI in different phenological stages. So we rewrite this part from Line 188 to Line 192 Now.

8. In Figure 4 next to Landsat also Modis and Sentinel-2 are mentioned: these sources were not used in this study? Better to remove from figure

Reply: yes, we only used the Landsat 7 ETM+ and Landsat 8 OLI image images, so we removed MODIS and Sentinel-2 now in Figure 4.

9. L189-191: from this description its not clear if you made a distinction between leaf and stalk points and if you removed the stalk points when calculating the leaf angle distribution. Keeping the stalk points will have an effect on the retrieval of the leaf angle distribution.

Reply: yes, indeed. So we removed the stalk points from leaves points when calculating the leaf angle distribution, which has been depicted in Line 245 to Line 253.

10. Table 3: the symbol / is used to indicate two issues: 1) landsat information; and 2) not relevant for example in case of one set value. Better to use two different symbols for this to avoid confusion. Which steps did you apply for the variable parameters to fill your LUT? What was the number of entries in your LUT?

Reply: yes, so we distinct these two issues now: symbol “/” represents values obtained from the Landsat data header, and “-” represents the one set value.

After the LUT dimension is determined, the sensitive input variables with setting range/value will be used to fill LUT, which has been depicted from Line 290 to Line 301.

There are about 100,000 spectra is loaded from the LUT library for LAI retrieval in this study, which has been depicted from Line 314 to Line 318.

11. L218-228: a few points are unclear in this paragraph: why do you only take into account the green and near-infrared band for the inversion. How do you combine the RMSE values for the two bands to make a selection for the best retrieval. Often model inversion approaches result in ill-posed solution: how did you deal with this for this study?

Reply: I am sorry for this error. We used four bands to retrieve LAI in this study, ranging from blue band to NIR band. And the blue band is omitted in previous version. The whole methodology has been rewritten, and the description about cost function is from Line 314 to Line 321.

The RMSE values are used to find the best match between simulated and observed reflectance from blue band to NIR band.

12. L238: probably you mean here insensitive instead of sensitive

Reply: this part is confused, so we have rewritten this paragraph now, which is from Line 217 to Line 228.

13. L249-250: the values indicated in the text differ from the values indicated in table 4. As you have repeated field measurements. To which extent are these values statistically different?

Reply: we are sorry for this confusion. The number in Table 4 is correct, and the numbers in text have been unified with Table 4 now.

14. Figure 6: titles and units for x and y-axis are missing, also statement of date is missing in the caption

Reply: yes, so we added the titles and units for x and y-axis in figure 4, and added the statement of date in the caption now.

15. L275: here you show the results for the September image: what about the other dates?

Reply: there is confusion in previous version, so we rewritten this part now. There are two groups of LAI retrieval validation. Firstly, the LAI retrieval results with default leaf angle and referred corn leaf angle distributions are compared with the extensive in situ measured LAIs on 4 September. This validation is done determine whether the introduction of inferred leaf angle distributions from TLS points data can improve the LAI retrieval accuracy using PROSAIL model. Figure 8 and Figure 9 is about this validation. Secondly, the LAI retrieval results on 10 July, 26 July, 19 August and 4 September with default and inferred corn leaf angle distribution functions are all compared with MODIS LAI in the whole study area. This comparation is done to evaluate the LAI retrieved results in a wide spatial range and temporal range. The Figure 10 is added for this validation, and the text is from Line 409 to Line 425.

16. Figure 8: titles and units for x and y-axis are missing

Reply: the titles and units for x and y-axis have been added now.

17. Figure 9: date is missing in caption: in L299 a different range is mentioned than is indicated in the legend to the detailed map

Reply: the date has been added in caption now. We are sorry for this confusion, 1.66 m2 /m2 to 2.45 m2 /m2 are the RMSEs between retrieved LAI and measured LAI in field work, and Figure 9 is the difference between LAIwith inferred LAD and LAIwith default LAD. So we have rewritten this part now, which is from Line 403 to Line 405.

18. L330: this result does not validate the improved LAI retrieval, rather is confirms the assumption that the leaf angle distribution is clearly changing during different phenological phases over the growing season

Reply: yes, it is. So we changed this sentence now.

19. L333-335: the current results only show the improvement (although not significant) for the observations in September as presented in Figure 6: the evaluation of the other dates is currently not presented and thus should be added to the results section to support this statement.

Reply: the comparation has been rewritten now, which is from Line 465 to Line 478.

20. L339-343: this part is a repetition from the results section: rather I would expect here a more elaborate assessment and explanation why the adapted leaf angle distribution would result in a better retrieval of LAI. Secondly, comparison with previous studies are currently completely lacking and need to be added to synthesise the results of this study with previous results. And third, the paper does not provide an outlook on how this result should be developed further. Lidar observations seem to be valuable but are often limited in availability. Can we derive reference values for the LAD of different crops according to fixed protocol of lidar measurements covering different crops around te world? This kind of elaboration would greatly contribute to improving the general applicability of RTMs for crop trait retrieval. Again also refer to previous studies as this work is not a stand alone research.

Reply: we have removed the repetition sentences about the numbers in accuracy assessment results. And the reason for adapting leaf angle distributions to result in a better LAI retrieval results is added from Line 465 to Line 470. The comparison with previous studies is added from Line 421 to Line 425, Line 475 to Line 478. The outlook on how this result should be developed further is added from Line 469 to Line 470, Line 484 to Line 487.

Round  2

Reviewer 2 Report

The authors have done a carefull job in improving the manuscript extending descriptions and providing better descriptions for unclear points. From my point of view the paper can be accepted.

This manuscript is a resubmission of an earlier submission. The following is a list of the peer review reports and author responses from that submission.